🔓 | Open Peer Review | Genetics and Molecular Biology | Observation

# Maximizing the potential of high-throughput next-generation sequencing through precise normalization based on read count distribution

Caitriona Brennan,[1] Rodolfo A. Salido,[2] Pedro Belda-Ferre,[1] MacKenzie Bryant,[1] Charles Cowart,[1] Maria D. Tiu,[3] Antonio González,[1] Daniel McDonald,[1] Caitlin Tribelhorn,[1] Amir Zarrinpar,[3,4,5] Rob Knight[1,2,4,6]

**ABSTRACT**   Next-generation sequencing technologies have enabled many advances across diverse areas of biology, with many benefiting from increased sample size. Although the cost of running next-generation sequencing instruments has dropped substantially over time, the cost of sample preparation methods has lagged behind. To counter this, researchers have adapted library miniaturization protocols and large sample pools to maximize the number of samples that can be prepared by a certain amount of reagents and sequenced in a single run. However, due to high variability of sample quality, over and underrepresentation of samples in a sequencing run has become a major issue in high-throughput sequencing. This leads to misinterpretation of results due to increased noise, and additional time and cost rerunning underrepresented samples. To overcome this problem, we present a normalization method that uses shallow iSeq sequencing to accurately inform pooling volumes based on read distribution. This method is superior to the widely used fluorometry methods, which cannot specifically target adapter-ligated molecules that contribute to sequencing output. Our normalization method not only quantifies adapter-ligated molecules but also allows normalization of feature space; for example, we can normalize to reads of interest such as non-ribosomal reads. As a result, this normalization method improves the efficiency of high-throughput next-generation sequencing by reducing noise and producing higher average reads per sample with more even sequencing depth.

**IMPORTANCE**   High-throughput next generation sequencing (NGS) has significantly contributed to the field of genomics; however, further improvements can maximize the potential of this important tool. Uneven sequencing of samples in a multiplexed run is a common issue that leads to unexpected extra costs or low-quality data. To mitigate this problem, we introduce a normalization method based on read counts rather than library concentration. This method allows for an even distribution of features of interest across samples, improving the statistical power of data sets and preventing the financial loss associated with resequencing libraries. This method optimizes NGS, which already has huge importance across many areas of biology.

**KEYWORDS**   metagenomics, large-scale studies, NGS normalization, automation, multiplexing, quantification, high-throughput sequencing

Metagenomic next-generation sequencing (metagenomic-NGS) is an increasingly useful tool in the field of biology and clinical medicine, allowing researchers to comprehensively sample all genes in all organisms present in a given complex sample. This tool enables microbiologists to evaluate bacterial diversity and detect the abundance of microbes in various environments. Importantly, it provides a means to study

Address correspondence to Rob Knight, robknight@eng.ucsd.edu.

The authors declare no conflict of interest.

See the funding table on p. 5.

unculturable microorganisms that are otherwise difficult or impossible to analyze. This has proven valuable to a whole range of scientific studies, such as microbiome characterization (1–3) pathogen detection (4) forensics (5), and environmental monitoring (6).

With advances in sequencing technology, the high number of reads provided by a single run on high-throughput sequencers such as the HiSeq or NovaSeq is driving the use of larger, multiplexed sample pools to lower sequencing costs (7–9). However, uneven library concentrations from different types and qualities of samples can lead to inconsistencies in data quality (10). Libraries with low concentration may be underrepresented on the flow cell, while those with high concentration are likely to be overrepresented. Overrepresentation can waste the finite data capacity of a sequencing run, whereas underrepresentation can lead to shallow read depth, unreliable data, and the squandering of valuable library material. Both cases lead to additional costs and time re-preparing libraries, which remain disproportionately high compared to per-base sequencing expenses (11). Additionally, decisions based on inaccurate or incomplete data could lead clinicians or researchers to miss critical information, making the choice of the right sequencing approach essential.

Normalization aims to mitigate these challenges ensuring every library is represented equally and sequenced to sufficient depths. To inform normalization, there are several options for quantitating library preps, which vary in ease and accuracy (10, 12). Spectrophotometry-based methods such as fluorometry, which are the quickest and most convenient, tend to be inaccurate (10). The accuracy of quantitation and subsequent normalization depends significantly on the quantification method's ability to detect adaptor-ligated double-stranded DNA molecules with specificity, which are the only molecules that can contribute to sequencing output. Since fluorometry cannot specifically target useful adaptor-ligated molecules, this is believed to result in the overestimation of the sequencing-competent library concentration. However, methods that can distinguish between adapter-ligated molecules, like quantitative PCR (qPCR), are time-consuming as they require knowing the average fragment size in each library for dilution calculations. To overcome these difficulties of metagenomic-NGS normalization, we optimized a method of normalization for hundreds of multiplexed samples that is based on read counts from a low-cost and rapid iSeq run (13).

We prepared shotgun metagenomic libraries from 352 samples plus 32 negative control extraction blanks according to our previously established protocol (14) using the HyperPlus library prep kit (KAPA Biosciences) (Text S1). As shown in Fig. S1, these libraries were quantified via the PicoGreen fluorescence assay (ThermoFisher, Inc) and pooled to approximately equal molar fractions using the Echo 550 robot. The resulting pool, representing 384 libraries, was sequenced on Illumina's iSeq, yielding a total combined depth of approximately 5 million paired-end reads. The read distribution of the 352 samples resulted in a normal distribution with each library occupying a median of 0.24% ±0.2% of the total reads per sample and a coefficient of variation of 0.72 (Fig. 1A). We created another pool where the pooled volume of each library was calculated based on the read distributions from this iSeq run (ranging from 10 nL to 1,000 nL), enabling normalization based on sequenced paired-end read counts (15). The iSeq normalized pool was sequenced on Illumina's iSeq to a total depth of approximately 5.7 million paired-end reads. Sequencing results from the iSeq normalized pool yielded a median library proportion of 0.3%±0.1% of the total reads per sample (in this case, ~16,000 ± 5,700 reads per sample) and a coefficient of variation of 0.37 (Fig. 1A). The significantly tighter standard deviation produced by this step demonstrates that hundreds of libraries can be pooled quickly and within close range of each other using this method. Despite over-penalizing some overrepresented samples, which caused some samples to underachieve the median read count (Fig. S2 and S3), this is a significant improvement in sequencing depth evenness across samples (Fig. 1B). Additionally, this normalization method allows us to normalize feature counts across samples, instead of raw read counts by changing the raw *reads PF$_i$* terms in the numerator and denominator

**(A)**

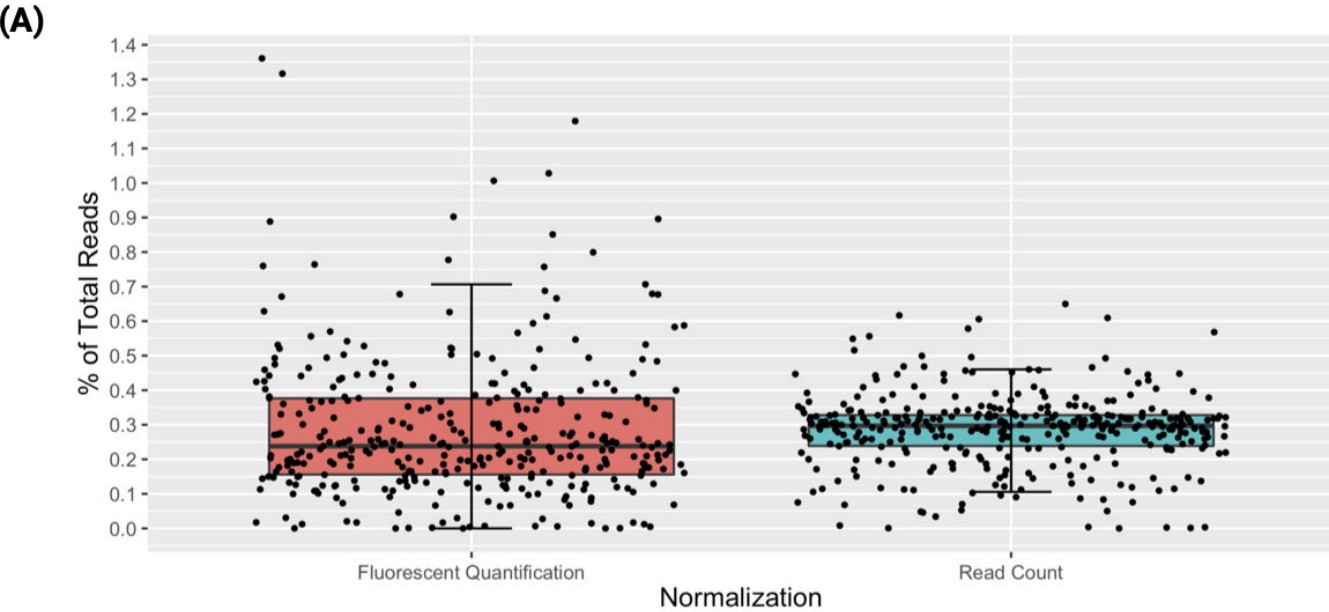

**(B)**

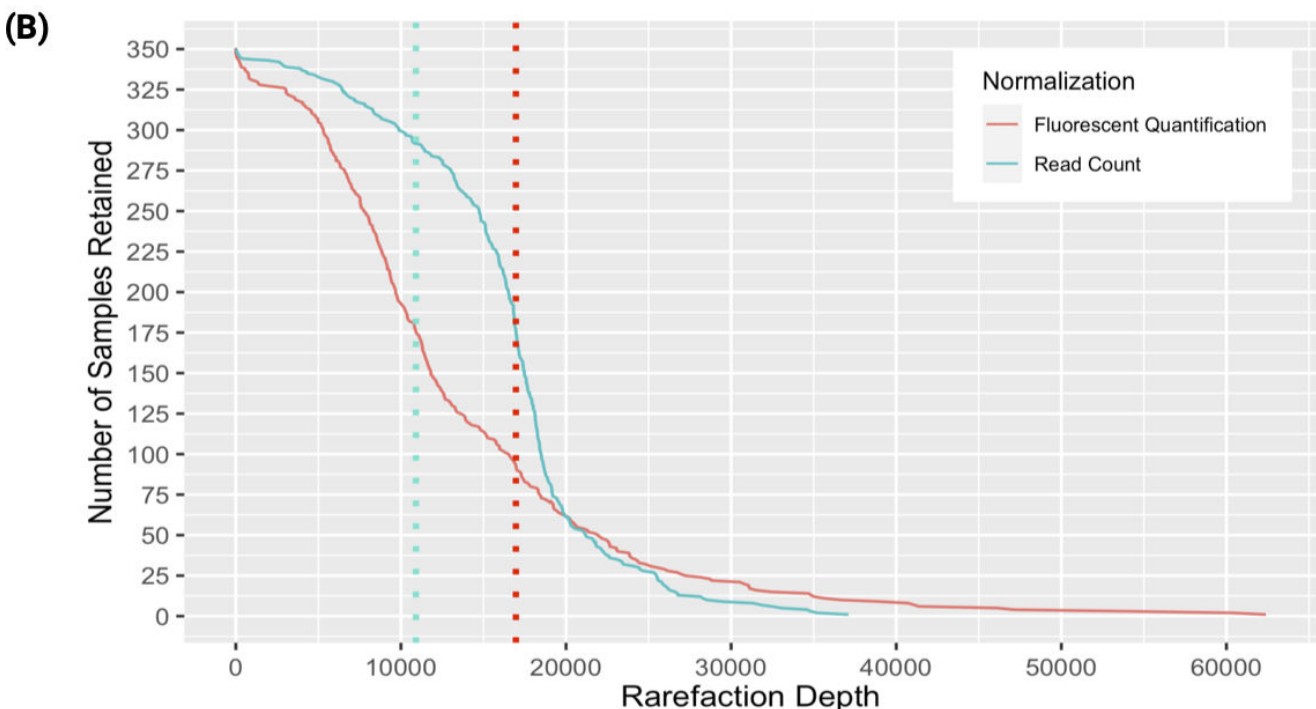

**FIG 1** (A) Boxplots showing the distribution of proportions of total reads observed in samples normalized with Fluorescent Quantification and Read Count normalization methods, respectively. Coefficient of variation for Fluorescent Quantification (pink/red) is 0.72. Coefficient of variation for Read Count (turquoise) is 0.37. (B) Rarefaction curve demonstrating the higher sample retention when rarefying to the median number of reads per normalization method. Dotted lines represent the median number of reads per sample for each method.

of the Reads%Index calculation (Fig. S1) for *feature counts$_i$* terms. For example, we can target reads of interest (features), such as non-ribosomal reads in metatranscriptomic sequencing (Fig. 2) or reads aligning to a specific genome for bacterial isolate sequencing, and normalize their counts across samples. This further reduces the amount of sequencing depth variation in the reads of interest for our downstream data analysis.

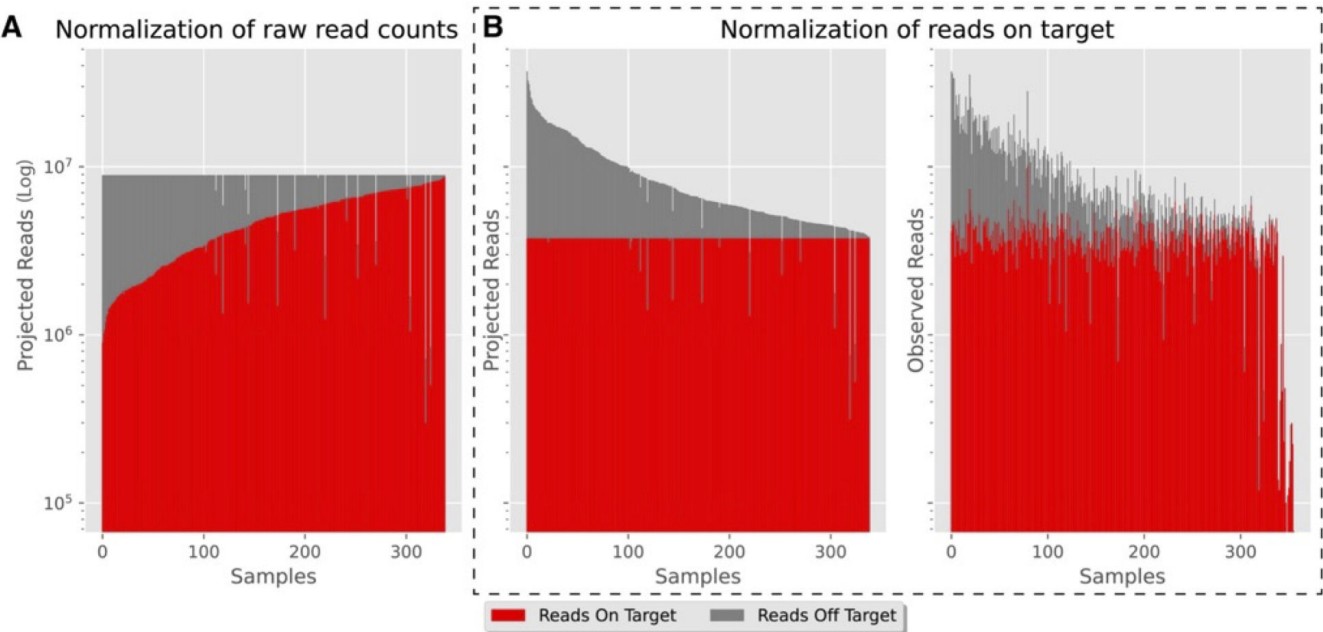

**FIG 2** Read count proportions of non-ribosomal reads (red, Reads On Target) for a metatranscriptomics sequencing project. (**A**) Projected read counts per sample with Read Count normalization of raw reads across samples. (**B**) Projected and observed read counts per sample with Read Count normalization of Reads On Target across samples.

The steps for preparing this additional sequencing pool include two fragment length distribution analyses, size selection, and quantification. As these steps are also required for preparing the final read count normalized pool, there are no additional capital costs, other than the iSeq. Further, the consumable costs are low when working with pooled samples (~$30 per pool). With personnel, it takes 1 technician approximately 6 h to prepare each pool for sequencing. Furthermore, it takes ~19 h (15–20 min hands-on time) and costs ~$500 to load and run an iSeq sequencer up to 8million paired-end reads. These costs are negligible in comparison to resequencing libraries on a Novaseq in order to make up for unsatisfactory normalization based on fluorescent quantification, where a typical Novaseq S4 run can take up to 44 h and costs between $4,000–$5,000 per lane. Moreover, the iSeq platform requires low input for a successful run, with a concentration of only 90 picomolar (pM) in 20 µL. This feature makes it feasible to use this read count normalization method with samples that have limited genetic material, such as skin swabs or other low biomass samples. QC steps, such as quantification and size selection, are performed on pooled samples; therefore, these steps also consume negligible amounts of each library. The deeper and more uniform sequencing produced from normalization by read distribution leads to higher sample retention when normalizing reads across samples with rarefaction and will allow for increased statistical power when testing for biological signals in a data set (Fig. 1B). Overall, the use of this normalization method will mitigate the risk of erroneous interpretation of results, improve identification and characterization of pathogenic organisms and microbial communities, and will also minimize the need to resequence libraries due to underrepresentation, saving time, and resources.

## ACKNOWLEDGMENTS

We thank Se Jin Song and Austin Swafford for their helpful discussions during the course of this work; Sawyer Farmer and Tara Schwartz for wet lab support; and Gail Ackermann for metadata support.

We thank Vienna Brunt, Nathan Greenberg, and principal investigator Prof. Douglas Seals, from the University of Colorado Boulder, for providing mouse fecal samples

supported by the National Institutes of Health (NIH) award, grant number R01 HL134887. A.Z. is supported by the R01 HL148801, R01 EB030134, and U01 CA265719. This work was also supported by the NIH Pioneer Award, grant number DP1AT010885 and the Alzheimer's Gut Microbiome Project, grant number U19AG063744. Authors received institutional support from NIH P30 DK120515 and UL1 TR001442.

## AUTHOR AFFILIATIONS

[1]Department of Pediatrics, University of California San Diego, La Jolla, California, USA

[2]Department of Bioengineering, University of California San Diego, La Jolla, California, USA

[3]Division of Gastroenterology, University of California San Diego, La Jolla, California, USA

[4]Center for Microbiome Innovation, University of California San Diego, La Jolla, California, USA

[5]VA San Diego Health Sciences, La Jolla, California, USA

[6]Department of Computer Science and Engineering, University of California San Diego, La Jolla, California, USA

## AUTHOR ORCIDs

Caitriona Brennan  http://orcid.org/0000-0003-3943-6701

Rodolfo A. Salido  http://orcid.org/0000-0003-0631-2817

Amir Zarrinpar  http://orcid.org/0000-0001-6423-5982

Rob Knight  http://orcid.org/0000-0002-0975-9019

## FUNDING

| Funder | Grant(s) | Author(s) |
| --- | --- | --- |
| HHS \| National Institutes of Health (NIH) | DP1AT010885 | Rob Knight |
| HHS \| NIH \| National Institute on Aging (NIA) | U19AG063744 | Rob Knight |
| HHS \| National Institutes of Health (NIH) | R01 HL148801, R01 EB030134, U01 CA265719, NIH P30 DK120515, UL1 TR001442 | Amir Zarrinpar |

## ADDITIONAL FILES

The following material is available online.

### Supplemental Material

**Figure S1 (mSystems00006-23-s0001.tif).** Flowchart of experimental design. 1. KAPA HyperPlus shotgun libraries are quantified using the PicoGreen fluorescence assay (ThermoFisher, Inc) and pooled to approximately equimolar fractions. 2. Pool is sequenced on Illumina's iSeq. 3. The resulting raw reads Passing Filter (PF) is used to calculate a Loading Factor for each library, which is the ratio between the index representing the highest proportion of the total reads PF and the index of each library's proportion of total reads PF (Illumina. [Internet]. 2019. Available from: https://www.illumina.com/content/dam/illumina-marketing/documents/systems/iseq/single-cell-library-qc-app-note-770-2019-029.PDF). This in turn scales the fluorescent quantified pooled volumes to calculate new pooling volumes. The new pooling volumes are clipped within a reasonable range for acoustic droplet ejection (typically between the range of 10 nL and 1,000 nL, using the Labcyte Echo 550). 4. Libraries are pooled using new pooling volumes. 5. The resulting read count normalized pool is sequenced on illumina's iSeq. Created with BioRender.com.

**Figure S2 (mSystems00006-23-s0002.tif).** Comparison of the top 5% of samples with the least and most reads for each normalization method. Dotted line represents the median percentage of total reads for both methods (Read Count normalized in turquoise and Fluorescent Quantification normalized in pink/red). (A) Top 5% of samples with the most amount of reads when Read Count normalization was applied. (B) Top 5% of samples with the most amount of reads when Fluorescent Quantification normalization was applied. (C) Top 5% of samples with the least amount of reads when Read Count normalization was applied. (D) Top 5% of samples with the least amount of reads when Fluorescent Quantification normalization was applied.

**Figure S3 (mSystems00006-23-s0003.tif).** Correlation between proportion of read counts per sample. Dotted lines represent the medians of each method. The diagonal line has a slope of 1. Some overrepresented samples from the Fluorescent Quantification Normalization method [% Total Reads {greater than or equal to} 3X (median % Total Reads )] were over penalized for the subsequent Read Count Normalization method, which resulted in a lower percentage of Total Reads than the median across samples within this normalization method. Nonetheless, the distribution of percentages of Total Reads across samples of the Read Count Normalization method was tighter and the median percentage of Total Reads across samples was higher.

**Text S1 (mSystems00006-23-s0004.docx).** Supplemental Materials and Methods.

## Open Peer Review

**PEER REVIEW HISTORY (review-history.pdf).** An accounting of the reviewer comments and feedback.

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
