## [Reviewer comments · mSystems]

Maximizing the potential of high-throughput next-generation sequencing through precise normalization based on read-count distribution

Caitriona Brennan, Rodolfo Salido, Pedro Belda-Ferre, MacKenzie Bryant, Charles Cowart, Maria Tiu, Antonio González, Daniel McDonald, Caitlin Tribelhorn, Amir Zarrinpar, and Rob Knight

Corresponding Author(s): Rob Knight, University of California San Diego

Review Timeline:

Submission Date:	January 5, 2023
Editorial Decision:	March 7, 2023
Revision Received:	March 27, 2023
Accepted:	April 24, 2023

Editor: Neha Sachdeva

Reviewer(s): The reviewers have opted to remain anonymous.

Transaction Report:

DOI: <https://doi.org/10.1128/msystems.00006-23>

March 7, 2023

Dr. Rob Knight
University of California, San Diego
Pediatrics
La Jolla, CA

Re: mSystems00006-23 (**Maximizing the potential of high-throughput next-generation sequencing through precise normalization based on read-count distribution**)

Dear Dr. Rob Knight:

Thank you for submitting your manuscript to mSystems. We have completed our review and I am pleased to inform you that, in principle, we expect to accept it for publication in mSystems. However, acceptance will not be final until you have adequately addressed the reviewer comments.

Preparing Revision Guidelines

Sincerely,

Neha Sachdeva

Editor, mSystems

Journals Department
American Society for Microbiology
1752 N St., NW

Reviewer comments:

Reviewer #1 (Comments for the Author):

This paper introduces a new method for NGS sample normalization based on read-counts. The new method offers a better way to track differences in metagenomes and metatranscriptomes amongst samples over fluorescent quantitation. As fluorescent quantitation will give a measurement of total sample, not necessarily sample that is amplified (and subsequently sequenced), this could provide increased confidence in comparing rarefied samples with limited material.

This paper is very well written. The only item I would consider changing is Fig.S2 so that Fluorescent quantification is on the same row and Read Count is on the same row, then alter legend accordingly.

Reviewer #2 (Comments for the Author):

This report from Brennan et al. demonstrates a new normalization method to correct for inaccuracies due to varying concentrations in the input volume. This work is significant in adding rigor to microbiome studies. There are a couple of questions that need further clarification.

While the method improves accuracy compared to qubit (most commonly used) in adjusting the input, what are the input volumes and cost (and resources) associated with running an iSeq first before the Novaseq run? Especially in low biomass critical patient samples, is this a feasible approach? Fecal pellets from mice have abundance of genomic material compared to patients samples such as swabs...

Can authors add more detail on the calculation for figure 2? How can this be applied to non-ribosomal reads? (or how can it be achieved)?

Reviewer comments:

Reviewer #1 (Comments for the Author):

This paper introduces a new method for NGS sample normalization based on read-counts. The new method offers a better way to track differences in metagenomes and metatranscriptomes amongst samples over fluorescent quantitation. As fluorescent quantitation will give a measurement of total sample, not necessarily sample that is amplified (and subsequently sequenced), this could provide increased confidence in comparing rarefied samples with limited material.

This paper is very well written. The only item I would consider changing is Fig.S2 so that Fluorescent quantification is on the same row and Read Count is on the same row, then alter legend accordingly.

Response: *Thank you for the kind words and the suggested improvement. We have updated the figure and legend with your suggestion.*

Reviewer #2 (Comments for the Author):

This report from Brennan et al. demonstrates a new normalization method to correct for inaccuracies due to varying concentrations in the input volume. This work is significant in adding rigor to microbiome studies. There are a couple of questions that need further clarification.

While the method improves accuracy compared to qubit (most commonly used) in adjusting the input, what are the input volumes and cost (and resources) associated with running an iSeq first before the Novaseq run? Especially in low biomass critical patient samples, is this a feasible approach? Fecal pellets from mice have abundance of genomic material compared to patients samples such as swabs...

Response: *Thank you for your comment, and raising these important questions. We have added the following to main text on lines 122 - 127 and 132 - 138, to address these points:*

“The steps for preparing this additional sequencing pool include two fragment length distribution analyses, size-selection, and quantification. As these steps are also required for preparing the final read count normalized pool, there are no additional capital costs, other than the iSeq. Further, the consumable costs are low when working with pooled

samples (~\$30 per pool). With personnel, it takes 1 technician approximately 6 hours to prepare each pool for sequencing.”

“ Moreover, the iSeq platform requires low input for a successful run, with a concentration of only 90 picomolar (pM) in 20 μ l. This feature makes it feasible to use this read count normalization method with samples that have limited genetic material, such as skin swabs or other low biomass samples. QC steps, such as quantification and size selection, are performed on pooled samples, therefore these steps also consume negligible amounts of each library.”

Can authors add more detail on the calculation for figure 2? How can this be applied to non-ribosomal reads? (or how can it be achieved)?

Response: *Thank you for your question. To normalize by feature space, for example by non-ribosomal reads, we used SortMeRNA (version v2.1b with default parameters) on adapter trimmed, raw reads passing filter (PF) to partition metatranscriptomic reads into ribosomal and non-ribosomal reads. The counts of non-ribosomal reads (reads on target, Fig. 2) replaced the raw reads PF_i terms in the numerator and denominator of the Reads%Index calculation (Fig. S1, #3).*

We have expanded on this in the main text (lines lines 114 - 121) and in the Materials and Methods section (Pooling and Sequencing).

April 24, 2023

Prof. Rob Knight
University of California San Diego
9500 Gilman Drive
MC 0602
La Jolla, CA 92093

Re: mSystems00006-23R1 (**Maximizing the potential of high-throughput next-generation sequencing through precise normalization based on read-count distribution**)

Dear Prof. Rob Knight:

Your manuscript has been accepted, and I am forwarding it to the ASM Journals Department for publication. For your reference, ASM Journals' address is given below. Before it can be scheduled for publication, your manuscript will be checked by the mSystems production staff to make sure that all elements meet the technical requirements for publication. They will contact you if anything needs to be revised before copyediting and production can begin. Otherwise, you will be notified when your proofs are ready to be viewed.

If you would like to submit a potential Featured Image, please email a file and a short legend to msystems@asmusa.org. Please note that we can only consider images that (i) the authors created or own and (ii) have not been previously published. By submitting, you agree that the image can be used under the same terms as the published article. File requirements: square dimensions (4" x 4"), 300 dpi resolution, RGB colorspace, TIF file format.

We recognize that the video files can become quite large, and so to avoid quality loss ASM suggests sending the video file via <https://www.wetransfer.com/>. When you have a final version of the video and the still ready to share, please send it to mSystems staff at msystems@asmusa.org.

Sincerely,

Neha Sachdeva
Editor, mSystems

Journals Department
E-mail: mSystems@asmusa.org